# Sequence-controlled supramolecular terpolymerization directed by specific molecular recognitions

Takehiro Hirao [1,2], Hiroaki Kudo[1], Tomoko Amimoto[3] & Takeharu Haino [1]

Nature precisely manipulates primary monomer sequences in biopolymers. In synthetic polymer sequences, this precision has been limited because of the lack of polymerization techniques for conventional polymer synthesis. Engineering the primary monomer sequence of a polymer main chain represents a considerable challenge in polymer science. Here, we report the development of sequence-controlled supramolecular terpolymerization via a self-sorting behavior among three sets of monomers possessing mismatched host–guest pairs. Complementary biscalix[5]arene-$C_{60}$, bisporphyrin-trinitrofluorenone (TNF), and Hamilton's bis(acetamidopyridinyl)isophthalamide-barbiturate hydrogen-bonding host–guest complexes are separately incorporated into heteroditopic monomers that then generate an ABC sequence-controlled supramolecular terpolymer. The polymeric nature of the supramolecular terpolymer is confirmed in both solution and solid states. Our synthetic methodology may pave an avenue for constructing polymers with tailored sequences that are associated with advanced functions.

[1] Department of Chemistry, Graduate School of Science, Hiroshima University, 1-3-1 Kagamiyama, Higashi-Hiroshima 739-8526, Japan. [2] Chemistry Department, NHB 5.340, The University of Texas at Austin, 100 24th St E, Austin, Texas 78712, USA. [3] Natural Science Center for Basic Research and Development, Hiroshima University, 1-3-1 Kagamiyama, Higashi-Hiroshima 739-8526, Japan. Correspondence and requests for materials should be addressed to T.H. (email: haino@hiroshima-u.ac.jp)

Structurally well-defined biopolymers are often found in nature. Proteins and nucleic acids achieve unique biological functions that are associated with their precisely defined backbone sequences. This exceptional sequence precision is not offered by synthetic polymers produced by conventional step-growth and chain-growth polymerizations. Therefore, scientific interest in establishing primary sequences of synthetic polymers has gained momentum for generating polymer materials[1–4]. Notably, a stepwise iterative synthesis[5] on a solid- or soluble-polymer support was developed, which ensures sequence-defined and monodisperse polymers with high batch-to-batch reproducibility[6–9]. Recent advances in sequence-regulated

polymers have employed sequence specificities in straightforward polymerization procedures. These specificities are determined by controlled polymerization conditions[10–15], monomer reactivities[16–21], and templates[22–24]. However, these strategies have difficulty generating polymers with perfectly sequence-specific microstructures. Step-growth polymerizations via acyclic diene metathesis polymerization[25], ring-opening metathesis polymerization[26, 27], or metal-catalyzed radical polymerization[28] are advantageous for employing elaborately designed monomers that possess tailored sequences for polymerization but lack control over the molecular weight and dispersity of the polymers. Although these methods produce repetitive sequences of

**Fig. 1** Supramolecular terpolymerization. **a** Schematic representation of the supramolecular terpolymerization of three components A, B, and C via self-sorting assembly or random assembly. **b** Host–guest structures of the biscalix[5]arene-C_{60} complex, bisporphyrin-TNF complex, and Hamilton's hydrogen-bonding complex. **c** Structures of the three heteroditopic monomers **1**, **2**, and **3** possessing mismatched host–guest pairs. **d** Sequence-controlled supramolecular polymerization of three sets of heteroditopic monomers via a self-sorting

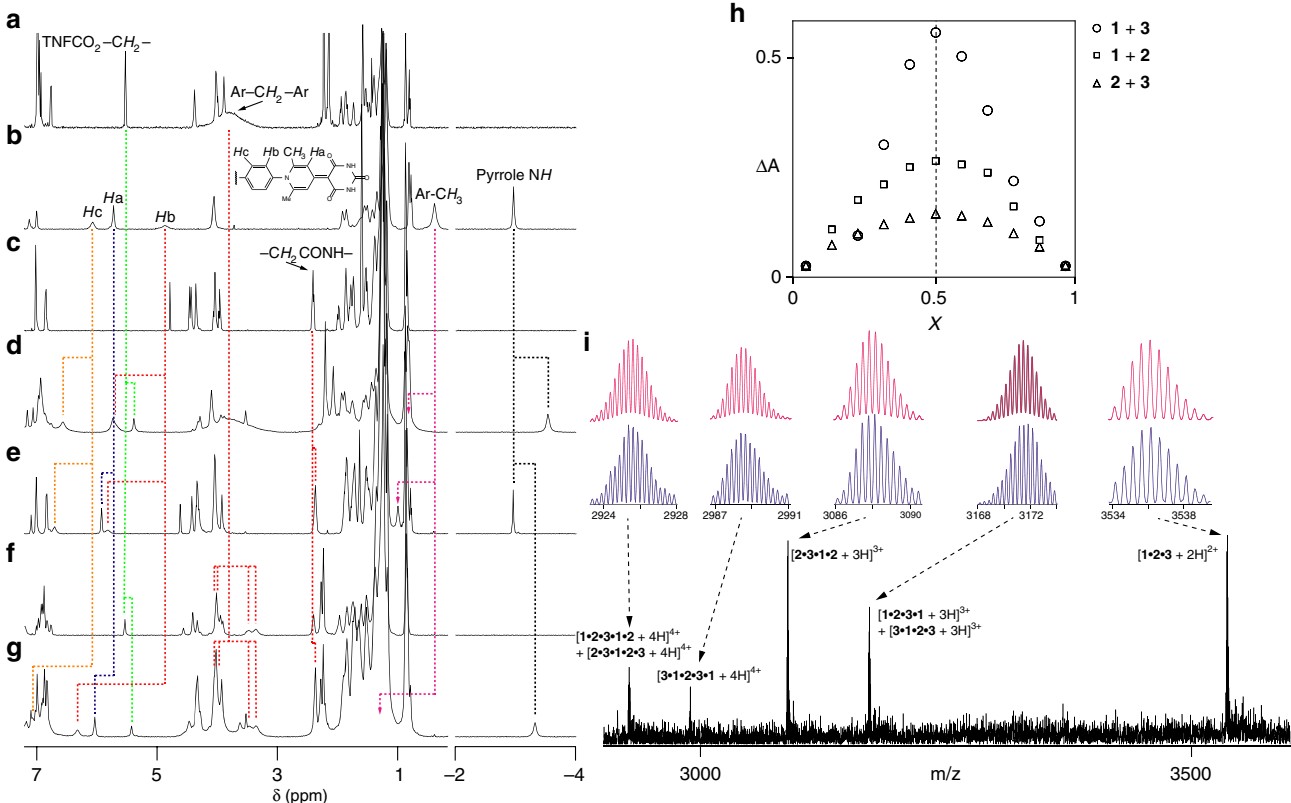

**Fig. 2** Self-sorting behaviors in supramolecular terpolymerization. The $^1$H NMR spectra of **a** **1**, **b** **2**, **c** **3**, **d** a mixture of **1** and **2**, **e** a mixture of **2** and **3**, **f** a mixture of **1** and **3**, and **g** a mixture of **1**, **2**, and **3** in chloroform-$d_1$. **h** Job plots for the **1•2**, **2•3**, and **1•3** heterodimeric complexes. **i** ESI-MS spectrum of a mixture of **1**, **2**, and **3**. Observed isotope distributions are highlighted in *blue*, and the calculated distributions are highlighted in *red*

monomers on the backbones of the resulting polymers, they can be technically considered homopolymerization processes.

Supramolecular polymerization can offer another approach for developing sequence-controlled polymer chains[29]. A self-sorting strategy was developed to regulate the sequence of monomer arrays in alternating copolymerization[30, 31]. To construct ABC sequences in supramolecular polymerization, high specificity is required for each binding event. Therefore, creating an ABC sequence-controlled supramolecular terpolymer is challenging via a self-sorting strategy (Fig. 1a). We developed unique host–guest motifs (i.e., a biscalix[5]arene-C$_{60}$ complex[32] and a bisporphyrin-trinitrofluorenone (TNF) complex[33]) that display high specificities in guest binding (Fig. 1b)[34–36]. Regulating the ABC sequence requires another host–guest interaction that displays high specificity to these host–guest complexes. A hydrogen-bonding complex between a Hamilton's host and a barbiturate fulfills this requirement[37]. We envisaged applying the biscalix[5]arene-C$_{60}$ complex, bisporphyrin-TNF complex, and Hamilton's hydrogen-bonding complex to develop a sequence-controlled supramolecular terpolymer in a self-sorting manner. Three heteroditopic monomers **1**, **2**, and **3** possessing mismatched pairs of host and guest moieties were designed (Fig. 1c). These monomers exhibit high specificity for each binding event. Therefore, the intermolecular associations can selectively result in specific dimers **1•2**, **2•3**, and **3•1** (Fig. 1d). This self-sorting behavior should determine the molecular array of $[1\text{–}2\text{–}3]_n$ in the sequence-controlled supramolecular terpolymer. Herein, we report the development of sequence-controlled supramolecular terpolymerization via a self-sorting behavior among three sets of heteroditopic monomers possessing mismatched host–guest pairs. The polymeric nature of the supramolecular terpolymer is confirmed in both the solution and solid states.

## Results

**Self-association**. The self-associations of monomers **1**, **2**, and **3** were investigated using $^1$H nuclear magnetic resonance (NMR) spectroscopy (Fig. 2a–c). The $^1$H NMR spectra of the monomers were slightly dependent on concentration, which gave association constants of $40 \pm 10$, $290 \pm 20$, and $3 \pm 2\,\mathrm{l\,mol^{-1}}$ for **1**, **2**, and **3**, respectively (Supplementary Figs. 1–6). The complexation-induced changes in chemical shift (CIS) for **1** and **3** were less than 0.1 p.p.m. and nonspecific, demonstrating that **1** and **3** formed random aggregates in chloroform (Supplementary Figs. 1, 5). In contrast, the self-assembly of **2** resulted in significant upfield shifts for the aromatic protons $H$a, $H$b, and $H$c as well as the Ar-C$H_3$ protons ($\Delta\delta = 0.66$, 2.76, 1.56, and 1.44 p.p.m.) (Fig. 2b and Supplementary Figs. 3, 7–9). These large CIS values place the electron-deficient barbiturate tail within the shielding region of the bisporphyrin cleft, resulting in head-to-tail oligomeric complexes (Supplementary Fig. 4).

**Self-sorting behavior**. Individual intermolecular associations between monomers **1**, **2**, and **3** were evaluated using $^1$H NMR spectroscopy. The intermolecular association between **1** and **2** resulted in the characteristic upfield shifts of the porphyrin N$H$ protons and the TNF COO-C$H_2$- methylene protons ($\Delta\delta = -0.57$ and $-0.22$ p.p.m.) (Figs. 2a, b, d and Supplementary Figs. 10–13), which indicates that the TNF moiety was selectively located within the bisporphyrin cleft. This host–guest complexation led to the simultaneous dissociation of self-assembled **2** with the downfield shift of the Ar-C$H_3$, $H$b, and $H$c protons ($\Delta\delta = 0.44$, 0.82, and 0.50 p.p.m.). Therefore, heterodimer **1•2** evidently formed in solution.

Adding monomer **3** to **2** gave rise to significant downfield shifts for the Ar-C$H_3$, $H$a, $H$b, and $H$c protons ($\Delta\delta = 0.61$, 0.20,

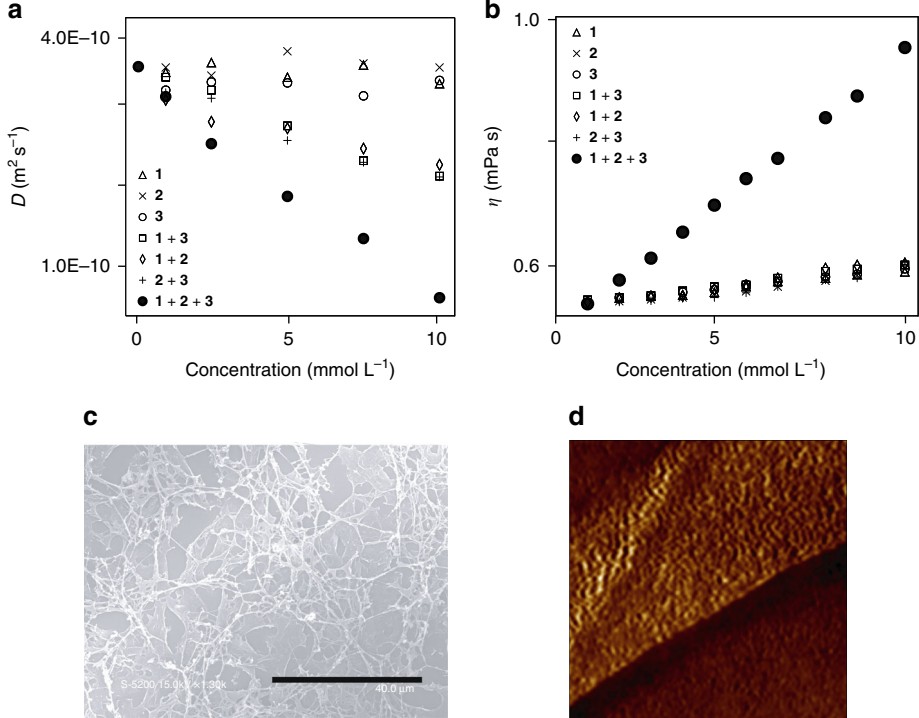

**Fig. 3** Properties and morphologies of supramolecular terpolymer [**1-2-3**]$_n$. **a** Diffusion coefficients (*D*) in chloroform-*d*$_1$. **b** Viscosities (*η*) in chloroform. **c** SEM image of supramolecular terpolymer. The *scale bar* denotes 40 μm. **d** AFM image (200 nm × 200 nm) of a spin-coated film

0.95, and 0.63 p.p.m.) (Figs. 2b, c, e and Supplementary Figs. 14–16) due to the dissociation of self-assembled **2**. The intermolecular nuclear Overhauser effect correlation between the *H*a proton of **2** and the -C*H*$_2$CONH- methylene protons of **3** indicated the close contact between the barbiturate tail of **2** and the Hamilton's host moiety of **3**, thus confirming the formation of the heterodimeric complex **2●3** (Supplementary Fig. 17).

A selective interaction between biscalix[5]arene and the C$_{60}$ moiety was clearly observed in the mixture of **1** and **3**. The broad Ar-C*H*$_2$-Ar resonance of **1** was due to the rapid ring flipping process of the calix[5]arene moieties (Fig. 2a). Upon encapsulating the C$_{60}$ moiety within the calix[5]arene cavities, the Ar-C*H*$_2$-Ar resonances split into two broad resonances, which clearly indicates that the ring-flipping process slows due to the attractive face-to-face contacts of the calix[5]arene interiors and the C$_{60}$ exterior (Fig. 2f and Supplementary Figs. 18–21). Therefore, the biscalix[5]arene-C$_{60}$ complexation directs the formation of heterodimeric **1●3**.

The stoichiometries and binding constants for the complementary host–guest pairs **1●2**, **2●3**, and **1●3** were determined using ultraviolet/visible (UV/vis) absorption spectroscopy (Fig. 2h). The **1●2**, **2●3**, and **1●3** host–guest complexes were each formed at a 1:1 ratio with high binding constants ($K_{1-2}$: 31,000 ± 1,000, $K_{2-3}$: 730,000 ± 5,000, and $K_{1-3}$: 15,000 ± 3,000 l mol$^{-1}$) (Supplementary Figs. 22–24). Consequently, the complementary host–guest pairs self-sort during the selective formation of heterodimeric pairs **1●2**, **2●3**, and **1●3** over the homodimers.

Finally, monomers **1**, **2**, and **3** were mixed in a 1:1:1 ratio, and the $^1$H NMR spectrum of the mixture was recorded. The same characteristics were observed for the assignable protons of Ar-C*H*$_2$-Ar, pyrrole N*H*, TNFCOO-C*H*$_2$-, Ar-C*H*$_3$, *H*a, *H*b, and *H*c (Fig. 2g and Supplementary Figs. 25–28,31). In particular, the large downfield shifts of the Ar-C*H*$_3$, *H*a, *H*b, and *H*c protons indicate the complete dissociation of self-assembled **2**. These results explicitly confirm that the biscalix[5]arene-C$_{60}$, bisporphyrin-TNF, and Hamilton's host–guest complexes exhibit self-

sorting in their intermolecular associations, which most likely results in the head-to-tail polymeric host–guest complex [**1–2–3**]$_n$ in sequence.

**Determination of the monomer sequence in the gas phase.** Electrospray ionization orbitrap mass spectrometry (ESI-MS) determines the constitutional repeating structures of the sequence-controlled supramolecular polymeric assemblies (Fig. 2i). In the ESI-MS spectra of the 1:1:1 mixture of **1**, **2**, and **3**, the heterodimeric pairs [**1●2**+3 H]$^{3+}$, [**2●3**+3 H]$^{3+}$, and [**3●1**+3 H]$^{3+}$ were primarily observed (Supplementary Figs. 32–36), suggesting that the supramolecular assembly among monomers **1**, **2**, and **3** exhibits self-sorting. Abundant peaks were observed at *m/z* = 2,926.6, 2,988.5, 3,088.1, 3,171.5, and 3,536.2, corresponding to [**1●2●3●1●2** + 4 H]$^{4+}$ + [**2●3●1●2●3** + 4 H]$^{4+}$, [**3●1●2●3●1** + 4 H]$^{4+}$, [**2●3●1●2** + 3 H]$^{3+}$, [**1●2●3●1** + 3 H]$^{3+}$ + [**3●1●2●3** + 3 H]$^{3+}$, [**1●2●3** + 2 H]$^{2+}$, respectively (Fig. 2i and Supplementary Figs. 37–41). These peaks were isotopically resolved and in good agreement with their calculated isotope distributions (Supplementary Tables 1–5). Therefore, the constitutional repeating structure **1–2–3** of the supramolecular polymer was confirmed in the gas phase. To exclude crossover repeating structures of **1-1-2-3**, **1-2-2-3**, and **1-2-3-3**, the supramolecular polymer was end-capped with competitive guests. C$_{60}$, 2,4,7-TNF, and 5-(*p*-methoxybenzylidene) barbituric acid (BA) completely dissociated the supramolecular polymeric assemblies, thus forming [C$_{60}$**●1●2** + 2 H]$^{2+}$, [C$_{60}$**●1●2●3** + 3 H]$^{3+}$, [2,4,7-TNF**●2** + 2 H]$^{2+}$, and [BA**●3** + 3 H]$^{3+}$ (Supplementary Figs. 42–48). Therefore, the repeating structure **1–2–3** in sequence was clearly established.

**Properties of the supramolecular terpolymer.** The formation of supramolecular polymers in solution was investigated using diffusion-ordered $^1$H NMR spectroscopy (DOSY). According to the Stokes–Einstein relationship, the diffusion coefficient (*D*) of a molecular species is inversely proportional to its hydrodynamic radius. The average size of a supramolecular polymer (*DP*) is

calculated based on the *D*s of existing molecular species. The *D*s of **1**, **2**, and **3** were independent of the concentration up to 10 mmol l$^{-1}$ (Supplementary Table 7). The **1**, **2**, and **3** exist in their monomeric form in solution (Fig. 3a). Although **2** exhibited self-assembly behavior in chloroform, the self-association was too weak to participate in the supramolecular homopolymerization. Upon concentrating the solutions from 1 to 10 mmol l$^{-1}$, the *D*s of **1** with **2**, **2** with **3**, and **3** with **1** reduced by ~35%, suggesting that these mixtures selectively formed heterodimeric complexes **1●2**, **2●3**, and **1●3**. In contrast, the *D*s of a 1:1:1 mixture of **1**, **2**, and **3** strongly depended on the concentrations. At 0.10 mmol l$^{-1}$, **1**, **2**, and **3** existed in their monomeric forms. Upon concentrating the solution to 10 mmol l$^{-1}$, the *D*s of the mixture gradually decreased by ~85%, which suggests that large polymeric aggregates were present. Assuming that the aggregates were hydro-spherical, an approximate *DP* of 200 was estimated at a concentration of 10 mmol l$^{-1}$.

Viscometry provides valuable information on the macroscopic size and structure of polymeric assemblies in solution. The solution viscosities of **1**, **2**, and **3** and their 1:1 mixtures were directly determined in chloroform (Fig. 3b and Supplementary Fig. 49 and Table 8). The solution viscosities of **1**, **2**, and **3** were not significantly influenced by their solution concentrations. When the solutions were concentrated, the viscosities of 1:1 mixtures of **1** and **2**, **2** and **3**, and **1** and **3** at 1:1 ratios did not meaningfully increase. Therefore, no polymeric aggregates formed. In contrast, a significant difference in the viscosity was observed for the 1:1:1 mixture of **1**, **2**, and **3**. As the solution concentrations increased, the solution became viscous, which suggests that well-developed polymer chains were formed at a concentration of ~10 mmol l$^{-1}$. Therefore, only the 1:1:1 mixture of **1**, **2**, and **3** resulted in the supramolecular polymeric chains that contribute to viscous drag.

Scanning electron microscopy (SEM) provided morphological insights into the supramolecular polymers in the solid state. The SEM images of cast films of **1**, **2**, and **3** and their 1:1 mixtures exhibited particle-like agglomerates (Supplementary Fig. 50). A 1:1:1 mixture of **1**, **2**, and **3** gave rise to polymeric fibers with diameters of 290 ± 50 nm, which partially formed sheet-like bundles (Fig. 3c)[38]. More detailed insight into the polymer formation was obtained using atomic force microscopy (AFM). Fig. 3d shows the uniform fibrous morphologies that were formed on the highly oriented pyrolytic graphite (HOPG) surface. The reasonably oriented fibrous morphologies possessed a uniform interchain distance of 3.9 ± 0.4 nm (Supplementary Figs. 51, 52 and Supplementary Table 9), which is consistent with the diameter of 3.6 nm calculated for the oligomeric structure (Supplementary Fig. 53).

## Discussion

In conclusion, we developed a ABC sequence-controlled supra-molecular terpolymer whose sequence is directed by employing the ball-and-socket, donor-acceptor, and hydrogen-bonding interactions that individually occur in the calix[5]arene-C$_{60}$, bisporphyrin-TNF, and Hamilton's complexes, respectively. The difference in the structural and electronic nature of these specific binding interactions evidently results in high-fidelity self-sorting, which provides control over the directionality and specificity in the sequence of the supramolecular terpolymer. Supramolecular chemistry offers various choices of host–guest motifs that have been previously developed with controllable structural and electronic properties. Therefore, our synthetic methodology may be extensively applied to the construction of tailored polymer sequences with structural variations and greater complexity by taking full advantage of host–guest motifs. Sequence-controlled supramolecular polymers developed using self-sorting are expected to provide possibilities for controlling advanced functions associated with polymer sequences, such as self-healing, stimuli responsiveness, and shape memory.

## Methods

**Characterization.** The characterization and synthesis of all compounds are described in full detail in the Supplementary Information. For the $^1$H NMR,$^{13}$C NMR, double-quantum filter correlation spectroscopy, heteronuclear single-quantum correlation spectroscopy, nuclear Overhauser spectroscopy, and ESI-Orbitrap mass spectra of the compounds in this article, see Supplementary Figs. 58–93.

**Determination of self-association constants for monomers.** The $^1$H NMR spectra of monomers **1**, **2**, and **3** were recorded at various millimolar concentrations in chloroform-$d_1$. Hyperbolic curves were obtained by plotting the compound concentrations as a function of the $^1$H NMR chemical shifts ($\delta$) of the aromatic protons of the monomers. The plots were fitted based on the isodesmic association model. The fitting functions are given by eq. (1), where $K$, $C$, $\delta_m$, and $\delta_a$ denote the association constant, total concentration of the compound, chemical shift for the monomer, and chemical shift of the self-assembled species, respectively (see Supplementary Figs. 1–6).

$$\delta(C) = \delta_m + (\delta_a - \delta_m)\left(1 + \frac{1 - \sqrt{4KC + 1}}{2KC}\right) \quad (1)$$

**Determination of host–guest stoichiometry.** A Job plot was used to determine the host–guest ratios for complexes **1●2**, **2●3**, and **3●1** in 1,2-dichloroethane at 25 ℃. A series of solutions containing two of the monomers were prepared such that the sum of the total concentrations of the monomers remained constant ($1 \times 10^{-5}$ mol L$^{-1}$). The mole fraction ($X$) was varied from 0.0 to 1.0. The absorbance changes ($\Delta A$) collected at 429 nm for **1●2**, at 450 nm for **2●3**, and at 470 nm for **3●1** were plotted as a function of the molar fraction.

**Determination of association constants.** A standard titration technique was applied for the determination of the association constants for the **1●2**, **2●3**, and **3●1** host–guest complexes in 1,2-dichloroethane at 25 ℃. A titration was performed wherein the concentration of a host solution ($1 \times 10^{-5}$ mol l$^{-1}$) was fixed while varying the concentration of its complementary guest. During the course of the titration, UV/vis absorption changes were measured from 250 to 900 nm. The experimental spectra were elaborated with the HypSpec program and subjected to a nonlinear global analysis by applying a 1:1 host–guest model of binding to determine the association constants (see Supplementary Figs. 22–24)[39].

**DOSY.** Monomers **1**, **2**, and **3** and their mixtures were dissolved in chloroform-$d_1$, and the sample solutions were placed in a 3 mm NMR sample tube. The pulse-field gradient diffusion NMR spectra were collected using a bipolar pulse pair stimulated echo (pulse sequence on a JEOL Delta 500 spectrometer with a 3 mm inverse H3X/FG probe[40]. The pulsed-field gradient strength was arrayed from ~0.003 to ~0.653 T m$^{-1}$ with a pulse gradient time of 1 ms and a diffusion time of 100 ms. The data were processed using the MestReNova program. The signal intensity as a function of the pulse-field gradient strength was fitted to the Stejskal–Tanner equation[41] to determine the diffusion coefficients.

**Solution viscosity measurements.** The solution viscosity of **1**, **2**, and **3** and their mixtures was measured at 25 ℃ with a rectangular slit m-VROC viscometer (RheoSense Inc.). Samples with different concentrations in chloroform were injected at flow rates of 0.11–0.28 ml min$^{-1}$ using a 0.5 ml syringe.

**SEM measurements.** Stock solutions of **1**, **2**, **3**, **1** with **2**, **2** with **3**, **3** with **1**, and **1** with **2** and **3** were prepared in 1,2-dichloroethane at concentrations of $2.0 \times 10^{-5}$ mol l$^{-1}$ with respect to each monomer. The stock solutions were drop-cast on a glass plate. The films were dried under reduced pressure for 9 h. A platinum coating was sputtered onto the films using a Hitachi Ion Sputter MC1000. The SEM images were recorded using a Hitachi S-5200 system.

**AFM measurements.** Stock solutions of **1**, **2**, **3**, **1** with **2**, **2** with **3**, **3** with **1**, and **1** with **2** and **3** were prepared in 1,2-dichloroethane at concentrations of $2.0 \times 10^{-5}$ mol l$^{-1}$ with respect to each monomer. The stock solutions were spin-coated onto freshly cleaved HOPG. The films were dried under reduced pressure for 9 h. The AFM measurements were performed using an Agilent 5100 microscope in air at ambient temperature with standard silicon cantilevers (NCH, NanoWorld, Neuchâtel, Switzerland), a resonance frequency of 320 kHz, and a force constant of 42 N m$^{-1}$ in tapping mode. The images were analyzed using the Pico Image processing program.

**ESI-MS measurements**. Stock solutions of **1**, **2**, and **3** were prepared in chloroform at concentrations of $2.8 \times 10^{-4}$ mol l$^{-1}$. Samples was prepared for ESI-MS by diluting by 15 times a mixture of 50 µl of each stock solution with a 2:1 mixture of chloroform and methanol, which was infused into the ESI source using a syringe pump at a flow rate of 5 µl min$^{-1}$ and analyzed using a spray voltage of 8 kV in positive ion mode. The ESI-MS measurements were performed using a Thermo Fisher Scientific LTQ Orbitrap XL hybrid FTMS system.

**Data availability**. The authors declare that the data supporting the findings of this study are available within the paper and its Supplementary Information File. All data are available from the authors on reasonable request.

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

## Acknowledgements

This work was supported by Grants-in-Aid for Scientific Research, JSPS KAKENHI Grant Numbers JP24350060 and JP15H03817, and by Grants-in-Aid for Scientific Research on Innovative Areas, JSPS KAKENHI Grant Numbers JP15H00946 (Stimuli-responsive Chemical Species), JP15H00752 (New Polymeric Materials Based on Element-Blocks), JP17H05375 (Coordination Asymmetry), and JP17H05159 (π-Figuration). T.Hirao thanks the Grant-in-Aid for JSPS Fellows, JSPS KAKENHI Grant Number JP13J02077. T.Haino thanks Daisuke Shimoyama for experimental assistance.

## Author contributions

T.Haino designed and conceived the study. T.Hirao and H.K. synthesized and measured all compounds. T.A. performed the ESI-MS measurements. T.Hirao, H.K., T.A., and T.Haino analyzed the data. T.Hirao, H.K., and T.Haino prepared the Supplementary Information, and T.Haino wrote the manuscript.

## Additional information

**Competing interests:** The authors declare no competing financial interests.

