## [Peer Review File · Nature Communications]

Reviewers' comments:

Reviewer #1 (Remarks to the Author):

This manuscript by Haino and coworkers describes the synthesis and characterization of periodic copolymers obtained by orthogonal self-assembly of supramolecular building blocks. In brief, three host-guest supramolecular combinations (i.e. biscalix[5]arene-fullerene, bisporphyrin-trinitrofluorenone and Hamilton's host/barbituric acid) were used to assemble three building blocks in a programmed sequence-controlled fashion. The supramolecular polymerization of these building blocks was studied in solution as well as in the solid-state and was characterized using NMR, ESI-MS, viscosity measurements, SEM and AFM. In particular, it was demonstrated that the targeted periodic sequence was obtained, although defects may occur. High-DP supramolecular polymerization was also confirmed by the experimental data. Thus, in my opinion, this manuscript could be of interest for the multidisciplinary readership of Nature Communications. Indeed, the concept is elegant and constitute an interesting step-forward in the field of supramolecular copolymers. The experimental study is also convincing. However, this manuscript requires in my opinion some adjustments:

1- The statement of the authors "This tremendous sequence precision has not been achieved for synthetic polymers produced by typical polymerization reactions" is not fully true anymore. The authors shall read the recent review of Lutz/Lehn/Meijer/Matyjaszewski (Nature Reviews Materials, 2016), in which modern polymerization tools are ranked. It clearly highlights that multi-step growth approaches allow today (almost) routine synthesis of sequence-defined polymers. The authors could cite, for example, recent examples reported by Alabi (JACS, 2014), Du Prez (JACS, 2016) and Lutz (Nature Communications, 2015). In particular, the latter group has reported preparation of sequence-defined polymers containing more than hundred residues (ACS Macro Letters, 2015).

2- The statement of the authors "However, these strategies have difficulty generating polymers with perfectly sequence-specific microstructures. More precise methods involve employing elaborately designed monomers that possess tailored sequences for polymerization via acyclic diene metathesis polymerization (ADMET), ring-opening metathesis polymerization (ROMP), or metal-catalyzed radical polymerization" is also incorrect. All the methods listed in the second sentence are step-growth procedures, which are rather unprecise. Indeed, they lead to polydisperse polymers (i.e. polydispersity index around 2) and are limited to repeating sequences (i.e. alternating or periodic). Thus, this section has to be slightly rephrased. On the other hand, the authors are correct in saying that these methods are homopolymerizations, while their concept is a self-sorting copolymerization.

3- Although the experimental work described in this manuscript is interesting from a fundamental point of view, the authors do not explain what properties and advances could be attained with their polymers. In particular, the conclusion of the article is short and does not contain clear perspectives. In the present work, the studied sequence is a model, in which the spacers of the three building blocks are quite similar (i.e. they do not carry specific molecular information). Besides this model, the authors shall describe what kind of functional sequences could be built using their approach.

Reviewer #2 (Remarks to the Author):

The manuscript by Haino and coworkers presents the supramolecular polymerization of bifunctional monomers via a self-sorting process. By design, the process affords sequence-controlled supramolecular polymers, as the recognition motifs are orthogonally designed. The heteroditopic monomers assemble via both hydrogen bonding and host-guest complexation based

upon Hamilton wedge (HW)-barbiturate, calix[5]arene-C60, and bisporphyrin-trinitrofluorenone (TNF) motifs, resulting in supramolecular ABC terpolymers.

The three ditopic monomers feature mismatched bis(calix[5]arene) and TNF (monomer 1), bisporphyrin and barbituric acid (monomer 2), and Hamilton wedge and C₆₀ (monomer 3) termini and are first examined for self-associations in solution. Monomers 1 and 3 did not display any significant induced chemical shifts via ¹H NMR spectroscopy, indicative of the formation of ill-defined aggregates. In contrast, 2 displayed significant shifting in of the porphyrin-related aromatic and benzyl signals, which are attributed to the complexation of the barbituric acid within the cleft of the bisporphyrin. Pairs of complementary monomers (e.g., 1 with 2, and 2 with 3) were then investigated using ¹H NMR spectroscopy, wherein bisporphyrin-TNF and HW-barbiturate pairs were clearly indicated by characteristic shifting of the porphyrin and HW aromatic resonances, respectively. In addition, the bis(calix[5]arene)-C₆₀ spectra depicted a face-to-face interaction of the bisporphyrin cleft and C₆₀ exterior.

Lastly, Haino and coworkers synthesized and characterized the 1:1:1 supramolecular terpolymer using NMR spectroscopy and mass spectrometry. The assembled mixture spectra displayed the same characteristics as observed for the complementary pairs (i.e., shifting of pyrrole NH, aromatic HW, calixarene aromatic resonances), while, importantly, no indications of self-associated 2 were observed. ESI-MS was also used to further characterize individual repeat units within the assembled mixture, wherein species indicative of the heterodimeric pairs were primarily found, owing to the self-sorting of the monomer units within solution. Additional species corresponding to 1-2-3-2-1 type arrangements were also observed. Further characterization of the assembly was conducted using viscometry and scanning electron microscopy. Viscometry measurements indicated the presence of well-defined polymer chains, as the solution becomes viscous using the 1:1:1 mixture, whereas viscometry measurements obtained at 1:1 mixtures were not appreciably viscous. SEM measurements also gave preliminary morphological insights, wherein fibrous networks were obtained.

The manuscript by Haino is both very interesting and timely, as multiple techniques are being established to attain sequence-control within polymer networks. The work detailed herein presents not only a simple system, but a simplistic route to obtain sequence-controlled supramolecular terpolymer networks and should be of interest to the readership of Nature Communications. It should be accepted after addressing some minor comments below.

- There is little doubt that the self-sorting behavior occurs, however, the authors have not included anywhere the full spectra related to barbiturate assemblies. As these protons show up downfield ranging from 9-13 ppm, the authors should also provide the full spectra to further show evidence of barbiturate-related assembly for both the self-association and terpolymer-related studies.
- The Hamilton wedge-barbiturate pair is a well-known hydrogen bonding motif. The authors call it host-guest recognition, which is accurate, but perhaps they should switch indications in text to hydrogen bonding, as that is the real interaction, as opposed to bisporphyrin-TNF, which is a complexation effect.
- Several references need to be re-formatted properly (missing page numbers, journal abbreviations, etc.)
- Solid state characterization using SEM of the supramolecular polymer appears very similar to the SEMs observed for PPVs. The author should consider referencing a paper or two.
- Why did the authors anneal to 80 C? Is this the expected T_g of the polymer or is this only for breaking the supramolecular bonds? The author should be careful of jumping to conclusions in the interpretation of the solid films. There is a lot of emerging literature addressing that supramolecular assembly in the solution state does not lend predictable phase separating

behavior. There is no evidence supporting whether or not polymer is supramolecularly bound before or after annealing in the solid state. This requires SAXS and WAXS, DSC, etc.

- The association constants as well as monomer stoichiometries were elucidated using UV-Vis spectroscopy. There needs to be more detailed explanation of this section as it is only touched upon in a couple sentences, but the SI shows multiple traces with little to no guidance of what the reader is looking at or looking for... the traces are very small making it hard to see and follow the data. At what wavelengths are you collecting the absorbance max(s) you are using to calculate the binding constants? Was this performed in triplicate? Have you done calibration curves of each ditopic monomer to ensure their absorbance increases linearly with increasing concentration (void of aggregation effects)?

Reviewer #3 (Remarks to the Author):

This ms reports a thorough piece of work done by an experienced investigator. The coding of molecular recognition elements that Haino has achieved goes far to offer an informational polymer that is a great distance from nucleic acids. The spectroscopic work is carefully done, and backs up the proposed guest conformations. This referee is not an expert in DOSY or SEM and hope the editors get advice from specialists. I expect the novelty of the work to be well-received by physical organic, supramolecular and spectroscopy chemists as Nature readers: I recommend acceptance. Before doing so I suggest the authors address the following points:

Remove the hype. The authors are overly fond of using "extremely" and "highly" appearing on p2.

Ref 24 should be to earlier work in self-sorting, informational polymers: J. Am. Chem. Soc. 1998, 120, 3657-3663.

Small amounts of acid (as provided by barbituric acid) would make 1 complementary to 2 and make 2 self-complementary. Can these interactions be excluded?

Point-by-point responses for the reviewers.

For Reviewer 1:

Comments: This manuscript by Haino and coworkers describes the synthesis and characterization of periodic copolymers obtained by orthogonal self-assembly of supramolecular building blocks. In brief, three host-guest supramolecular combinations (i.e. biscalix[5]arene-fullerene, bisporphyrin-trinitrofluorenone and Hamilton's host/barbituric acid) were used to assemble three building blocks in a programmed sequence-controlled fashion. The supramolecular polymerization of these building blocks was studied in solution as well as in the solid-state and was characterized using NMR, ESI-MS, viscosity measurements, SEM and AFM. In particular, it was demonstrated that the targeted periodic sequence was obtained, although defects may occur. High-DP supramolecular polymerization was also confirmed by the experimental data. Thus, in my opinion, this manuscript could be of interest for the multidisciplinary readership of Nature Communications. Indeed, the concept is elegant and constitute an interesting step-forward in the field of supramolecular copolymers. The experimental study is also convincing. However, this manuscript requires in my opinion some adjustments:

Thank you very much for the encouraging comments.

1- The statement of the authors "This tremendous sequence precision has not been achieved for synthetic polymers produced by typical polymerization reactions" is not fully true anymore. The authors shall read the recent review of Lutz/Lehn/Meijer/Matyjaszewski (Nature Reviews Materials, 2016), in which modern polymerization tools are ranked. It clearly highlights that multi-step growth approaches allow today (almost) routine synthesis of sequence-defined polymers. The authors could cite, for example, recent examples reported by Alabi (JACS, 2014), Du Prez (JACS, 2016) and Lutz (Nature Communications, 2015). In particular, the latter group has reported preparation of sequence-defined polymers containing more than hundred residues (ACS Macro Letters, 2015).

Thank you very much for the instructive comments. I have learnt the recent advances in precision polymer synthesis from the review article indicated by the reviewer. I found some oversights in terms of supported synthesis. To highlight supported synthesis, I made a slight modification in the main text.

The sentence in P2, "This tremendous sequence precision has not been achieved for synthetic polymers produced by typical polymerization reactions. Therefore, scientific interest in

establishing primary sequences of synthetic polymers has gained momentum for generating novel polymer materials.¹⁻³,”

was revised to be

“This exceptional sequence precision is not offered by synthetic polymers produced by conventional step-growth and chain-growth polymerizations. Therefore, scientific interest in establishing primary sequences of synthetic polymers has gained momentum for generating novel polymer materials.¹⁻⁴ Notably, a stepwise iterative synthesis⁵ on a solid- or soluble-polymer support was developed, which ensures sequence-defined and monodisperse polymers with high batch-to-batch reproducibility.⁶⁻⁹”

The references indicated by the reviewer, Alabi (JACS, 2014), (Lutz ACS Macro Letters, 2015), Lutz (Nature Commun., 2015) and Du Prez (JACS, 2016) were cited as references 6-9. The reference, Lutz/Lehn/Meijer/Matyjaszewski (Nature Reviews Materials, 2016) was also cited as reference 4. In addition, a reference dealing with Merrifield’s solid phase synthesis was cited as reference 5.

2- The statement of the authors "However, these strategies have difficulty generating polymers with perfectly sequence-specific microstructures. More precise methods involve employing elaborately designed monomers that possess tailored sequences for polymerization via acyclic diene metathesis polymerization (ADMET), ring-opening metathesis polymerization (ROMP), or metal-catalyzed radical polymerization" is also incorrect. All the methods listed in the second sentence are step-growth procedures, which are rather unprecise. Indeed, they lead to polydisperse polymers (i.e. polydispersity index around 2) and are limited to repeating sequences (i.e. alternating or periodic). Thus, this section has to be slightly rephrased. On the other hand, the authors are correct in saying that these methods are homopolymerizations, while their concept is a self-sorting copolymerization.

Thank you for the comments. I agree with the reviewer. I revised the sentence pointed out by the reviewer to make sure these reaction conditions are classified to be step-growth procedures.

The sentence,

“More precise methods involve employing elaborately designed monomers that possess tailored sequences for polymerization via acyclic diene metathesis polymerization (ADMET), ring-opening metathesis polymerization (ROMP), or metal-catalyzed radical polymerization,”

was changed to be

“Step-growth polymerizations via acyclic diene metathesis polymerization (ADMET),²⁵ ring-opening metathesis polymerization (ROMP),^{26,27} or metal-catalyzed radical polymerization²⁸ are advantageous for employing elaborately designed monomers that possess tailored sequences for polymerization but lack control over the molecular weight and dispersity of the polymers.”

3- Although the experimental work described in this manuscript is interesting from a fundamental point of view, the authors do not explain what properties and advances could be attained with their polymers. In particular, the conclusion of the article is short and does not contain clear perspectives. In the present work, the studied sequence is a model, in which the spacers of the three building blocks are quite similar (i.e. they do not carry specific molecular information). Besides this model, the authors shall describe what kind of functional sequences could be built using their approach.

I agree with the review. In line with the reviewer's comment, I revised the last part of this manuscript. The last sentence, “Our achieved construction of a sequence-controlled supramolecular terpolymer via self-sorting provides invaluable insight for the further development of polymers with greater complexity. The self-sorting supramolecular polymerization reported may prove beneficial when exploring applications of polymers in advanced materials.” was removed. New sentences describing the conclusion and perspective of the concept were added: “In conclusion, we developed a novel ABC-sequence-controlled supramolecular terpolymer whose sequence is directed by employing the ball-and-socket, donor-acceptor, and hydrogen-bonding interactions that individually occur in the calix[5]arene-C₆₀, bisporphyrin-TNF, and Hamilton's complexes, respectively. The difference in the structural and electronic nature of these specific binding interactions evidently results in high-fidelity self-sorting, which provides control over the directionality and specificity in the sequence of the supramolecular terpolymer. Supramolecular chemistry offers various choices of host-guest motifs that have been previously developed with controllable structural and electronic properties. Therefore, our synthetic methodology may be extensively applied to the construction of tailored polymer sequences with novel structural variations and greater complexity by taking full advantage of host-guest motifs. Sequence-controlled supramolecular polymers developed using self-sorting are expected to provide new possibilities for controlling advanced functions associated with polymer sequences, such as self-healing, stimuli responsiveness, and shape memory.”

For Reviewer 2:

Comments: The manuscript by Haino and coworkers presents the supramolecular polymerization of bifunctional monomers via a self-sorting process. By design, the process affords sequence-controlled supramolecular polymers, as the recognition motifs are orthogonally designed. The heteroditopic monomers assemble via both hydrogen bonding and host-guest complexation based upon Hamilton wedge (HW)-barbiturate, calix[5]arene-C60, and bisporphyrin-trinitrofluorenone (TNF) motifs, resulting in supramolecular ABC terpolymers.

The three ditopic monomers feature mismatched bis(calix[5]arene) and TNF (monomer 1), bisporphyrin and barbituric acid (monomer 2), and Hamilton wedge and C-60 (monomer 3) termini and are first examined for self-associations in solution. Monomers 1 and 3 did not display any significant induced chemical shifts via ^1H NMR spectroscopy, indicative of the formation of ill-defined aggregates. In contrast, 2 displayed significant shifting in of the porphyrin-related aromatic and benzyl signals, which are attributed to the complexation of the barbituric acid within the cleft of the bisporphyrin. Pairs of complementary monomers (e.g., 1 with 2, and 2 with 3) were then investigated using ^1H NMR spectroscopy, wherein bisporphyrin-TNF and HW-barbiturate pairs were clearly indicated by characteristic shifting of the porphyrin and HW aromatic resonances, respectively. In addition, the bis(calix[5]arene)-C60 spectra depicted a face-to-face interaction of the bisporphyrin cleft and C60 exterior.

Lastly, Haino and coworkers synthesized and characterized the 1:1:1 supramolecular terpolymer using NMR spectroscopy and mass spectrometry. The assembled mixture spectra displayed the same characteristics as observed for the complementary pairs (i.e., shifting of pyrrole NH, aromatic HW, calixarene aromatic resonances), while, importantly, no indications of self-associated 2 were observed. ESI-MS was also used to further characterize individual repeat units within the assembled mixture, wherein species indicative of the heterodimeric pairs were primarily found, owing to the self-sorting of the monomer units within solution. Additional species corresponding to 1-2-3-2-1 type arrangements were also observed. Further characterization of the assembly was conducted using viscometry and scanning electron microscopy. Viscometry measurements indicated the presence of well-defined polymer chains, as the solution becomes viscous using the 1:1:1 mixture, whereas viscometry measurements obtained at 1:1 mixtures were not appreciably viscous. SEM measurements also gave preliminary morphological insights, wherein fibrous networks were obtained.

The manuscript by Haino is both very interesting and timely, as multiple techniques are being established to attain sequence-control within polymer networks. The work detailed herein presents not only a simple system, but a simplistic route to obtain sequence-controlled supramolecular terpolymer networks and should be of interest to the readership of Nature Communications. It should be accepted after addressing some minor comments below.

Thank you very much for the encouraging comments.

- There is little doubt that the self-sorting behavior occurs, however, the authors have not included anywhere the full spectra related to barbiturate assemblies. As these protons show up downfield ranging from 9-13 ppm, the authors should also provide the full spectra to further show evidence of barbiturate-related assembly for both the self-association and terpolymer-related studies.

Thank you for the comments. The ¹H NMR spectra of 1, 2, 3, and their mixtures were already shown in Supplementary Figure 18. As seen in Figure 18f,g, there are the broad protons found downfield ranging from 9 ppm to 13 ppm. Although some of them could be assignable to the hydrogen-bonded NH protons of the HW-barbiturate complex, we failed to provide the reliable assignment for the hydrogen-bonded protons due to the severer overlaps of the NH protons of 2 and 3. However, the intermolecular NOE between the HW and the barbiturate tail in the NOESY spectrum (Supplementary Figure 86) confirmed that the intermolecular hydrogen-bonded the HW-barbiturate complex between 2 and 3 was formed. To clarify the NOE assignment, we revised the Supplementary Figure 86.

- The Hamilton wedge-barbiturate pair is a well-known hydrogen bonding motif. The authors call it host-guest recognition, which is accurate, but perhaps they should switch indications in text to hydrogen bonding, as that is the real interaction, as opposed to bisporphyrin-TNF, which is a complexation effect.

Thank you for the constructive suggestion. I agree with it. In the main text, we made the following changes:

In p2, the sentence, "Hamilton's bis(acetamidopyridinyl)isophthalamide-barbituric acid host-guest," was revised to be "Hamilton's bis(acetamidopyridinyl)isophthalamide-barbiturate hydrogen-bonding host-guest."

In p3, the sentence, "A Hamilton's host-guest complex fulfills this requirement.," was revise to be "A hydrogen-bonding complex between a Hamilton's host and a barbiturate fulfills this requirement."

In p4., the sentence, "biscalix[5]arene-C₆₀, bisporphyrin-TNF, and Hamilton's host-guest complexes," was revised to be "biscalix[5]arene-C₆₀ complex, bisporphyrin-TNF complex, and Hamilton's hydrogen-bonding complex."

In Figure 1 legend, "Hamilton's host-barbituric acid complex" was revised to "Hamilton's hydrogen-bonding complex."

- Several references need to be re-formatted properly (missing page numbers, journal abbreviations, etc.)

Thank you very much for pointing out our oversights. We carefully corrected them.

- Solid state characterization using SEM of the supramolecular polymer appears very similar to the SEMs observed for PPVs. The author should consider referencing a paper or two.

Thank you for the comment. I looked through papers reporting PPV films. I found a reference (*Materials Letters* **2007**, 61, 2159) dealing with the SEM image of PPV polymers. I hope this fits what the reviewer indicated. We cited the reference as reference 38.

- Why did the authors anneal to 80 C? Is this the expected Tg of the polymer or is this only for breaking the supramolecular bonds? The author should be careful of jumping to conclusions in the interpretation of the solid films. There is a lot of emerging literature addressing that supramolecular assembly in the solution state does not lend predictable phase separating behavior. There is no evidence supporting whether or not polymer is supramolecularly bound before or after annealing in the solid state. This requires SAXS and WAXS, DSC, etc.

Thank you very much for the reviewer's comments. Temperature- and solvent-annealing techniques are often applied to prepare a uniform supramolecular organization on a HOPG substrate to obtain a high-quality AFM image. We carried out a temperature annealing of a spin-coated film of the supramolecular polymer at 80°C which is lower than decomposition temperatures of 150°C and 165°C for 1 and 3, and mp of >300°C for 2. A film prepared on a glass plate instead of a HOPG substrate was subjected to X-ray diffraction analysis. There was no diffraction in the scanning angle of 2θ ranging from 5° to 80°, which indicated that the film

was amorphous. We didn't have any further result dealing with the structure of the supramolecular organization of the annealed film on HOPG as the reviewer pointed out. I think of it is premature that the polymeric structures are still maintained after the annealing. Therefore, we moved the image Figure 3E into Supplementary Figure 41, and removed the sentence, "Upon annealing at 80 °C, the fibrous morphologies became a uniform film with a thickness of 3.21 nm (Fig. 3E). Therefore, the spread supramolecular polymer chains were reorganized to produce a bundle of polymers due to the noncovalent interchain interactions on the substrate."

- The association constants as well as monomer stoichiometries were elucidated using UV-Vis spectroscopy. There needs to be more detailed explanation of this section as it is only touched upon in a couple sentences, but the SI shows multiple traces with little to no guidance of what the reader is looking at or looking for... the traces are very small making it hard to see and follow the data. At what wavelengths are you collecting the absorbance max(s) you are using to calculate the binding constants? Was this performed in triplicate? Have you done calibration curves of each ditopic monomer to ensure their absorbance increases linearly with increasing concentration (void of aggregation effects)?

Thank you for pointing out our oversight. The intermolecular host-guest complexations of 1•2, 2•3, and 3•1 were evaluated using UV/vis titration. The spectral changes of a host observed in a range from 250 nm to 900 nm were all collected upon the addition of its guest. Hypspec program used all of the changes observed in the range to determine the association constants. To clarify how to do the titrations, we added the following sentences in the methods section:

"Determination of host-guest stoichiometry. A Job plot was used to determine the host-guest ratios for complexes 1•2, 2•3, and 3•1 in 1,2-dichloroethane at 25°C. A series of solutions containing two of the monomers were prepared such that the sum of the total concentrations of the monomers remained constant (1×10^{-5} mol L⁻¹). The mole fraction (X) was varied from 0.0 to 1.0. The absorbance changes collected at 429 nm for 1•2, at 450 nm for 2•3, and at 470 nm for 3•1 were plotted as a function of the molar fraction.

Determination of association constants. A standard titration technique was applied for the determination of the association constants for the 1•2, 2•3, and 3•1 host-guest complexes in 1,2-dichloroethane at 25°C. A titration was performed wherein the concentration of a host

solution (1×10^{-5} mol L⁻¹) was fixed while varying the concentration of its complementary guest. During the course of the titration, UV/vis absorption changes were measured from 250 nm to 900 nm. The experimental spectra were elaborated with the HypSpec program and subjected to a non-linear global analysis by applying a 1:1 host-guest model of binding to determine the association constants (see Supplementary Figs. 15–17).³⁹

We cited a new reference (Gans, P., Sabatini, A. & Vacca, A. *Talanta* **43**, 1739-1753, (1996)) that describes the fitting program as ref 39.

To make sure the self-associations of the monomers are negligible in the concentration range applied for the titration, the calibration curves of the monomers were prepared (See Supplementary Figure 14a). The good linear correlations indicate that **1**, **2**, and **3** exist as monomeric forms in the millimolar concentrations, in which we carried out the host-guest titrations.

For Reviewer 3:

Comments: This ms reports a thorough piece of work done by an experienced investigator. The coding of molecular recognition elements that Haino has achieved goes far to offer an informational polymer that is a great distance from nucleic acids. The spectroscopic work is carefully done, and backs up the proposed guest conformations. This referee is not an expert in DOSY or SEM and hope the editors get advice from specialists. I expect the novelty of the work to be well-received by physical organic, supramolecular and spectroscopy chemists as Nature readers: I recommend acceptance.

Before doing so I suggest the authors address the following points:

We really appreciate the review's encouraging comments.

Remove the hype. The authors are overly fond of using "extremely" and "highly" appearing on p2.

We removed two "extremely," and one "highly," which appeared in p2.

Ref 24 should be to earlier work in self-sorting, informational polymers: J. Am. Chem. Soc. 1998, 120,

3657-3663.

Thank you for indicating the paper reporting an outstanding earlier work. We cited the paper suggested as reference 30.

Small amounts of acid (as provided by barbituric acid) would make 1 complementary to 2 and make 2 self-complementary. Can these interactions be excluded?

The reviewer concerns that barbituric acid would react with some basic nitrogens of 1 and 2, influencing the intermolecular association of 1 and 2. Substituted barbituric acids such as diethyl barbiturate are weak acids, which give a pKa of approximately 7.4. I think of the barbiturate moiety of 2 should be less acidic than non-substituted barbituric acid; therefore, it should not react with the nitrogens of 1 and 2. To make sure the protonated structures of 1 and 2 don't exist in solution, we added an excess amount of non-substituted barbituric acid (pKa = 4.0) into solutions of 1 and 2, and then UV/vis spectra were measured (See Supplementary Figure 14b,c). The addition of barbituric acid didn't result in any significant difference in absorption, suggesting that the triazole moiety and the porphyrin rings are not influenced with the barbiturate moiety of 2 in solution. We added the absorption spectra of 1 and 2 in the presence and in the absence of the barbituric acid in Supplementary Figure 14b,c.

REVIEWERS' COMMENTS:

Reviewer #1 (Remarks to the Author):

The authors have carefully revised their manuscript. In particular, they have addressed all the comments that I raised. Their answers are convincing and I therefore recommend publication in Nature Communications.

Reviewer #2 (Remarks to the Author):

Reviewer #2 only made comments to the editor and is satisfied with the revised manuscript.

Reviewer #3 (Remarks to the Author):

I AM SATISFIED THAT THE AUTHORS HAVE PROPERLY RESPONDED TO THE CRITIQUES IN THE REVISION.

THE MS IS NOW OK FOR PUBLICATION.

Point-by-point responses for the reviewers.

Reviewer #1 (Remarks to the Author):

The authors have carefully revised their manuscript. In particular, they have addressed all the comments that I raised. Their answers are convincing and I therefore recommend publication in Nature Communications.

Reviewer #2 (Remarks to the Author):

Reviewer #2 only made comments to the editor and is satisfied with the revised manuscript.

Reviewer #3 (Remarks to the Author):

I AM SATISFIED THAT THE AUTHORS HAVE PROPERLY RESPONDED TO THE CRITIQUES IN THE REVISION.

THE MS IS NOW OK FOR PUBLICATION.

Thank you very much for providing very valuable comments. I have learnt our manuscript is now publishable in *Nature Communications*.